# Effects of *Cistanche tubulosa* Wight Extract on Locomotive Syndrome: A Placebo-Controlled, Randomized, Double-Blind Study

**DOI:** 10.3390/nu13010264

**Published:** 2021-01-18

**Authors:** Yuna Inada, Chihiro Tohda, Ximeng Yang

**Affiliations:** Section of Neuromedical Science, Institute of Natural Medicine, University of Toyama, Toyama 930-0194, Japan; yuna@inm.u-toyama.ac.jp (Y.I.); ximeng@inm.u-toyama.ac.jp (X.Y.)

**Keywords:** locomotive syndrome, *Cistanche tubulosa*, walking ability, two-step test, 5-m walking

## Abstract

In an aging society, preventing dysfunction and restoring function of the locomotive organs are necessary for long-term quality of life. Few interventional studies have investigated supplementation for locomotive syndrome. Additionally, very few interventional clinical studies on locomotive syndrome have been performed as placebo-controlled, randomized, double-blind studies. We previously found that the administration of 30% ethanolic extract of *Cistanche tubulosa* improved walking ability in a cast-immobilized skeletal muscle atrophy mouse model. Therefore, we conducted a clinical study to evaluate the effects of *C. tubulosa* (CT) extract on the locomotive syndrome. Twenty-six subjects with pre-symptomatic or mild locomotive syndrome completed all tests and were analyzed in the study. Analyses of muscle mass and physical activity were performed based on the full analysis set. Intake of CT extract for 12 weeks increased step width (two-step test) and gait speed (5 m walking test) in patients over 60 years old compared with those in a placebo control (*p* = 0.046). In contrast, the skeletal muscle mass of the body trunk and limbs was unchanged following administration of CT extract. Adverse effects were evaluated by blood tests; no obvious adverse events were observed following the intake of CT extract. In conclusion, this placebo-controlled, randomized, double-blind study demonstrated that treatment with CT extract significantly prevented a decline in walking ability without any notable adverse effects in patients with locomotive syndrome.

## 1. Introduction

In an aging society, preventing dysfunction and restoring function of the locomotive organs are necessary for long-term quality of life. Locomotive syndrome was first described by the Japanese Orthopedic Association (JOA) in 2007, and it encompasses a wider range of disabilities than musculoskeletal ambulation disability symptoms [1]. The locomotive system includes muscles, joints, cartilage, and bones that gradually weaken with age or due to other diseases. Factors, such as lack of exercise, a sedentary lifestyle, and inadequate nutrition, also contribute to the progression of locomotive syndrome. Prevention and amelioration of the locomotive syndrome are required to reduce the population in need of care. Most interventional studies on the locomotive syndrome have been performed to evaluate the beneficial effects of exercise. For example, exercise for 3 months based on locomotive training significantly improved physical function tests [2]. Step length, stride length, and walking speed improved in patients with locomotive syndrome following 6-weeks of hipflexor muscle training [3]. However, most interventional studies on the effects of exercise have compared pre- and post-exercise responses in the same patients, without the inclusion of a control group. In addition, few interventional studies have investigated supplementation for locomotive syndrome; vitamin D supplementation for 24 weeks significantly improved the strength of knee extension and hip flexion compared with pre-treatment levels [4]. However, the study did not include a placebo group [4]. To date, a few interventional clinical trials on locomotive syndrome have been performed as placebo-controlled, randomized, double-blind studies [5].

Previously, we showed that administration of 30% ethanolic extract of *Cistanche tubulosa* improved walking ability in a cast-immobilized skeletal muscle atrophy mouse model [6]. Although oral administration of the extract did not increase skeletal muscle mass or myofiber diameter, walking skill and gait speed improved [6]. Cistanche herb is defined in the Japanese Pharmacopoeia as fleshy stems of *Cistanche salsa* G.Beck, *C. deserticola* Y.C.Ma, or *C. tubulosa* Wight. Acteoside (synonym: verbascoside), and echinacoside are recognized as the major active constituents of this plant. Acteoside- and echinacoside-rich *C. deserticola* extract has demonstrated antifatigue activity in mice [7]. Acteoside pre-treatment significantly increased muscle contractility in Bufo gastrocnemius, indicating an anti-muscle fatigue effect of this compound [8]. In our previous study, acteoside treatment significantly improved motor function in mice with spinal cord injuries [9]. Cistanche herb is classified as a non-pharmaceutical agent by the Pharmaceutical and Food Safety Bureau, Ministry of Health, Labour, and Welfare in Japan. The safety of *C. tubulosa* (CT) extract in humans has been established in phase II and phase III clinical studies performed in China. Those studies reported no adverse effects at a dose of 1800 mg extract per day for 3 months [10]. Therefore, we conducted a clinical study to evaluate the ameliorative effect of CT extract on locomotive syndrome in human adults. We hypothesized that CT extract might protect a decline of motor function.

## 2. Materials and Methods

### 2.1. Trial Design

This was placebo-controlled, randomized, double-blind study in healthy adults. The subjects applied for the study by conducting their own locomotive checkup after seeing a flyer inviting them to participate in the study placed at a nearby facility. After agreeing to participate in the study at the University of Toyama, the subjects underwent locomotive testing at baseline. The details of the locomotive clinical examination will be described later. Then, in the intervention phase, subjects were given a randomized drug (CT extract or placebo), and subjects one stick daily for 12-week at home. After the 12-week intervention, the subjects underwent locomotive testing again.

### 2.2. Participants

The prior power analysis was conducted. Calculated using the Wilcoxon matched-pairs signed rank test with an effect size of 0.5, significance level of 0.05, and power of 0.8, and then the required sample size was 28. Thirty-two subjects were recruited between 1 December 2018 and 31 October 2019. Potential subjects were allocated into two groups; 28 met the inclusion criteria and were enrolled. Two subjects voluntarily withdrew, thus 26 subjects participated in the study. All subjects visited the University of Toyama twice for assessments. A Consolidated Standards of Reporting Trials (CONSORT) diagram for the study is shown in Figure 1. The inclusion criteria were as follows: subjects (a) aged ≥40 and ≤80 years; (b) able to complete this clinical study; and (c) checked at least one item on the “Loco-check” questionnaire. The exclusion criteria were as follows: subjects (a) aged <39 years; (b) pregnant or lactating females; (c) on medication, such as muscle relaxants, osteoporosis, or rheumatoid arthritis drugs; (d) diagnosed with mental illnesses; or (e) judged ineligible for other reasons. Subjects were followed up from 11 May 2019 to 29 February 2020. This study was conducted with the approval of the Ethics Committee of the University of Toyama (R2018090). Each subject signed an informed consent form prior to study entry.

### 2.3. Intervention

Placebo or CT was given to each subject at a study venue on the first experiment day. The CT extract was prepared by Alps Pharmaceutical Ind. Co., Ltd. (Hida, Japan) as follows: *C. tubulosa* fleshy stems were collected from Shinjang Uyghur Aptonom Rayoni, People’s Republic of China. Mixed powders (20 kg) of *C. tubulosa* were immersed in 30% ethanol. The mixture was refluxed for 2 h at 55–64 °C. The extract yield was 21.5% and contained 5.41% acteoside and 17.48% echinacoside.

Safety assessments on the other lots of CT extract (acteoside < 9.0%; echinacoside < 25.0%) were performed by Oryza Oil & Fat Chemical Co., Ltd. The CT extract presented no acute toxicity in mice (LD50 > 26,400 mg/kg), no chronic toxicity in rats (at 1650 mg/kg after 180 days of administration), and no evidence of gene mutagenesis. Clinical safety assessments revealed no adverse effects during 180-days’ administration at 1800 mg/subject/day [10].

One stick contained 1800 mg of CT extract, 72 mg fine silicon dioxide, and 1782 mg dextrin. One stick, once per day, was taken with swallowing aid jelly (Ryukakusan Co., Ltd., Tokyo, Japan). Placebo powder contained 48 mg caramel coloring, 72 mg fine silicon dioxide, and 3480 mg dextrin. One stick was taken once daily with swallowing aid jelly. Mixing powders and packaging were prepared by the manufacturer (Sankyo Co., Ltd., Fuji, Japan) following Good Manufacturing Practice controls and ISO 22,000 certification. The dose of CT extract taken each day was 1800 mg.

### 2.4. Outcomes and Assessments

All participants completed a basic sociodemographic and medical history questionnaire and reported any medications used at baseline. Subjects who checked at least one item on the Loco-check questionnaire [11], given below, were recruited and considered to be at risk of locomotive syndrome:You cannot put on a pair of socks while standing on one leg.You stumble or slip in your house.You need to use a handrail when going upstairs.You cannot get across the road at a crossing before the traffic light changes.You have difficulty walking continuously for 15 min.You find it difficult to walk home carrying a shopping bag weighing approximately 2 kg.You find it difficult to do housework requiring physical strength.

Assessments were conducted in groups of one to ten participants per day, depending on the availability of the participants. All items were administered in one day, and participation was about 30 min. The order of measurements started with blood collection, height measurement, and grip strength measurement, while other items were carried out individually and randomly. During the participation, subjects were allowed to take a break whenever they wanted.

#### 2.4.1. Measurement of Muscle Mass

Upper and lower limb muscle mass and body trunk muscle mass were determined via bioelectrical impedance analysis with a body composition monitor (MC-780A, Tanita, Tokyo, Japan).

#### 2.4.2. Hand Grip Strength

Hand grip strength was measured using a hand dynamometer (T.K.K.5001, Takei, Niigata, Japan).

#### 2.4.3. Five-Meter Walking Speed

“Five-meter walking speed” measures the time that passes during a 5 m walking section by setting acceleration sections of 1.0 m at the start of a total 6 m of walking length. A start point, 1.0 m and finish points were marked by lines. A date collector judged and measured time using a stopwatch. Measurements were performed twice, and the fastest walking speed, without running, was recorded.

#### 2.4.4. Two-Step Test

The two-step test assesses walking ability [12]. The subject starts from a standing posture and is asked to take steps forward with maximum stride without losing balance. A specific mat with scales for the two-step test (JOA) was used. The length of two steps was measured from toe to toe. Before performing the test, an instructor demonstrated stepping. The score is calculated using the total length of two steps (cm) divided by the subject’s height (cm).

#### 2.4.5. The Stand-Up Test

The stand-up test is used to measure lower limb muscle strength. It evaluates an individual’s ability to stand using both legs, initially, and then on one leg, from a sitting position on stools at heights of 40, 30, 20, and 10 cm [13]. The results were expressed as summed scores, using the conversion Table 1.

#### 2.4.6. GLFS-25

The 25-question Geriatric Locomotive Function Scale (GLFS-25) is a self-rated questionnaire used to evaluate locomotor function in elderly individuals [14]. It includes 25 items with a score of 0–4 for each item: four questions regarding pain, 16 questions regarding daily activities, three questions regarding social function, and two questions regarding mental health status. The score ranges from 0 to 100, with higher scores indicating a poorer condition.

### 2.5. Safety Assessment

Safety assessments included the recording of adverse events and biochemical blood tests to assess liver and renal function, and blood sugar, and lipid levels at each visit.

### 2.6. Randomization

Participants were randomly assigned to one of two groups: the CT extract group or the placebo group. Randomization was performed by a simple randomization method by a third party who secured the participant allocation list and performed a key opening.

### 2.7. Statistical Analysis

The results were expressed as the mean ± standard deviation (SD). A 95% confidence interval (CI) denotes the interval that is 95% certain to contain the true population value, as it might be estimated from a much larger study. The 95% CI limits are shown by the lower and upper ranges. Statistical comparisons were performed using GraphPad Prism 6 (GraphPad Software, La Jolla, CA, USA) and SPSS (IBM, Chicago, IL, USA). Based on the results of the two tests of normality (Kolmogorov–Smirnov test and Shapiro–Wilk test) and the values of skewness and kurtosis, it was decided to use nonparametric tests for all statistical analyses. Data were analyzed using the Mann Whitney test (for intergroup comparison), Wilcoxon matched-pairs signed rank test (for intragroup comparison), or Fisher’s exact test (for number of people comparison). *p* values < 0.05 were considered significant.

## 3. Results

### 3.1. Baseline Characteristics of the Study Groups

The study population comprised 32 males and females. Figure 1 presents the CONSORT flow diagram, subject distribution, and individual study protocols. Thirty-two subjects were randomized into two groups. Four subjects were excluded because they did not meet the inclusion criteria. One group was allocated to receive placebo for 12 weeks, and the other group was allocated to receive CT extract for 12 weeks. Twenty-six subjects completed all tests and were included in the subsequent analyses; baseline characteristics are shown in Table 2. The interventions were evaluated in the subjects with pre-symptomatic or mild locomotive syndrome. Analyses of muscle mass and physical activity were performed on the full analysis set. The age of subjects in the CT extract group was significantly higher than that of subjects in the placebo group. Height, body weight, initial body-mass index (BMI), and initial Locomo 7 scores were not different between the placebo and CT extract groups (Table 2).

### 3.2. Muscle Mass and Physical Activities

There were no significant differences in the muscle mass of the body trunk, arms, and legs, or body weight between the placebo and CT extract groups (Table 3). Intragroup comparison pre- and post-treatment with placebo or CT extract revealed no significant changes in any of the above variables (Table 3). Hand grip, 5 m walking speed, two-step, and stand-up test scores, and GLFS-25 score were evaluated to assess physiological activity. Inter- and intragroup comparisons (Table 4, Figure 2A) revealed no significant differences in the outcomes of any tests.

Next, physical activities were analyzed by stratifying by age (>60 or >65 years). The improvement in the two-step test score in the CT extract group was significantly larger than that in the placebo group in subjects aged >60 years (Table 5, Figure 2B). A more evident positive effect of CT extract on the two-step test score was observed in subjects aged >65 years (Table 6, Figure 2C). Intragroup comparisons indicated that the two-step test score worsened post-treatment in the placebo group whereas that in the CT extract group was maintained during the test period (Table 6). No significant differences in muscle mass and body weight were found, even when stratified by age.

The number of subjects aged >60 years whose scores improved in both the 5 m walking and two-step test was compared between groups. The intake of CT extract significantly improved the chances of a positive response in the subjects (Figure 3).

### 3.3. Safety

Thirty-one parameters in blood, as well as adverse events were evaluated following the intake of CT extract and placebo. The results of the safety evaluation are listed in Table 7. Interestingly, sodium ion leucine aminopeptidase levels were significantly changed between the placebo and CT extract groups. No other parameters differed significantly between groups. Raw sodium data in the placebo group pre- and post-treatment were 140.8 and 142.0 mEq/L, respectively. In contrast, raw sodium data in the CT extract group pre- and post-treatment were 141.7 and 141.7 mEq/L, respectively. These findings indicate that sodium ion levels were not affected by the intake of CT extract, and that they were within the normal range in both the groups (more than 136 mEq/L). Raw data for leucine aminopeptidase in the placebo group pre- and post-treatment were 56.7 and 62.0 U/L, respectively. In contrast, raw data for leucine aminopeptidase in the CT extract group pre- and post-treatment were 54.1 and 53.5 U/L, respectively. These values indicate that there was no change in leucine aminopeptidase levels following CT extract intake, and leucine aminopeptidase levels in both groups were within the normal range (37–81 U/L). Overall, no adverse effects were observed following the intake of CT extract.

## 4. Discussion

This study is the first to report that 12-weeks’ intake of CT extract can increase step width (two-step test) and gait speed (5 m walk test) in patients aged over 60 years compared with a placebo control. Despite the age of CT group was older than that of placebo group (Table 2), CT intake significantly improved those physical activities. This result indicates the effect of CT extract is probably reliable. In contrast, the skeletal muscle masses of the body trunk and limbs did not change by CT extract intake. We used the same CT extract in a previous animal study. Oral administration of the CT extract for 13 days improved walking ability in cast-immobilized muscle atrophy mice, but did not increase skeletal muscle mass [6]. Several human studies have reported that muscle mass and muscle quality, such as strength, are not necessarily synchronized. A large observational study, including 1880 aged subjects, showed that muscle strength declined during 36-months follow-up without any decline in muscle mass [15]. Another controlled interventional study showed that resistance training for 6 months significantly increased muscle strength of the lower extremities; however, skeletal muscle mass did not change by the intervention [16].

Although step width in the placebo group decreased 12 weeks after in Figure 2B,C, the step width in the CT extract group was maintained. A natural decline in step width over a few months was observed in another study. A study of healthy adults (average age: 57 years) indicated that the two-step test score declined gradually within 8, 12, and 16 weeks even in the placebo-treated group [17]. Taken together, the data indicate that CT extract may protect against the attenuation of muscle strength. Several interventions have been investigated in humans to enhance walking performance, including vitamin D supplements [4] and milk proteins, such as ricotta [18], whey [19], and amino acids [19,20,21]. A significant improvement in gait speed was demonstrated in one study [22], but not in others [4,18,19,20,21]. Also in mice, branched-chain amino acid supplementation improved physical activity [23]. Although a positive effect of exercise on walking performance has been reported, additive improvement with amino acid supplementation was negatively evaluated [24]. Therefore, the number of potential supplementation candidates that protect against locomotive syndrome is limited. The results of the present study indicated that CT extract can safely prevent a decline in walking performance, suggesting that CT extract might be a novel agent for locomotive syndrome.

Regarding the mechanism underlying the effect of CT extract on muscle strength, we hypothesize that this involves enhanced axonal innervation from neurons to skeletal muscle by acteoside, an active constituent in CT extract. We previously identified acteoside as the main active constituent in CT extract (data not shown) and showed that it can activate axonal elongation activity, resulting in up-regulation of motor function [9]. However, the precise molecular mechanism underlying the effect of CT extract or acteoside on muscle strength needs to be analyzed using appropriate animal models in the future.

Several limitations of this investigation should be noted. We enrolled a sample of Asian adults aged 40–80 years; therefore, our results cannot be extended to other populations. Furthermore, we did not assess daily dietary intake or physical activity level; it is, therefore, unclear whether these impacted the study results. The participants were advised not to modify their eating habits or activity patterns during the intervention. Finally, the study was limited by the small sample size.

## 5. Conclusions

In conclusion, this placebo-controlled, randomized, double-blind study revealed that treatment with CT extract significantly prevented a decline in walking ability without any adverse effects. Based on the evidence in this study, we will promote the development of CT extract as Foods with Functional Claims and Over-The-Counter medicine in Japan.

## Figures and Tables

**Figure 1 nutrients-13-00264-f001:**
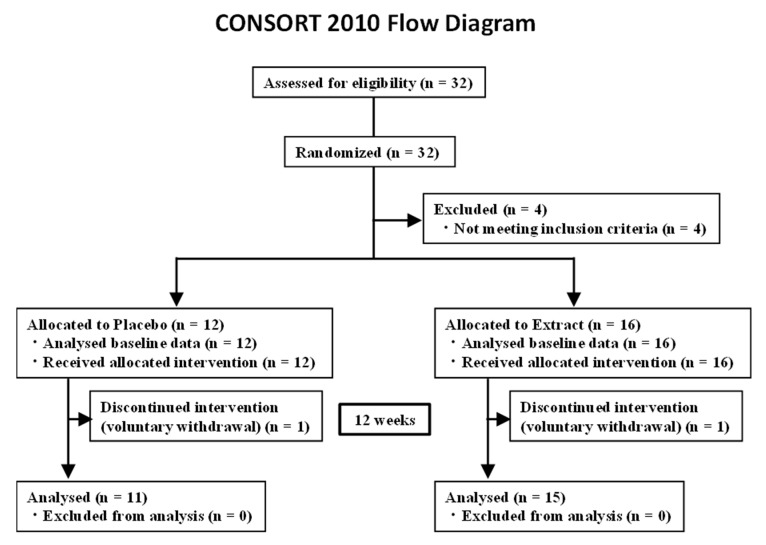
Study flow (CONSORT 2010 diagram). CONSORT, Consolidated Standards of Reporting Trials.

**Figure 2 nutrients-13-00264-f002:**
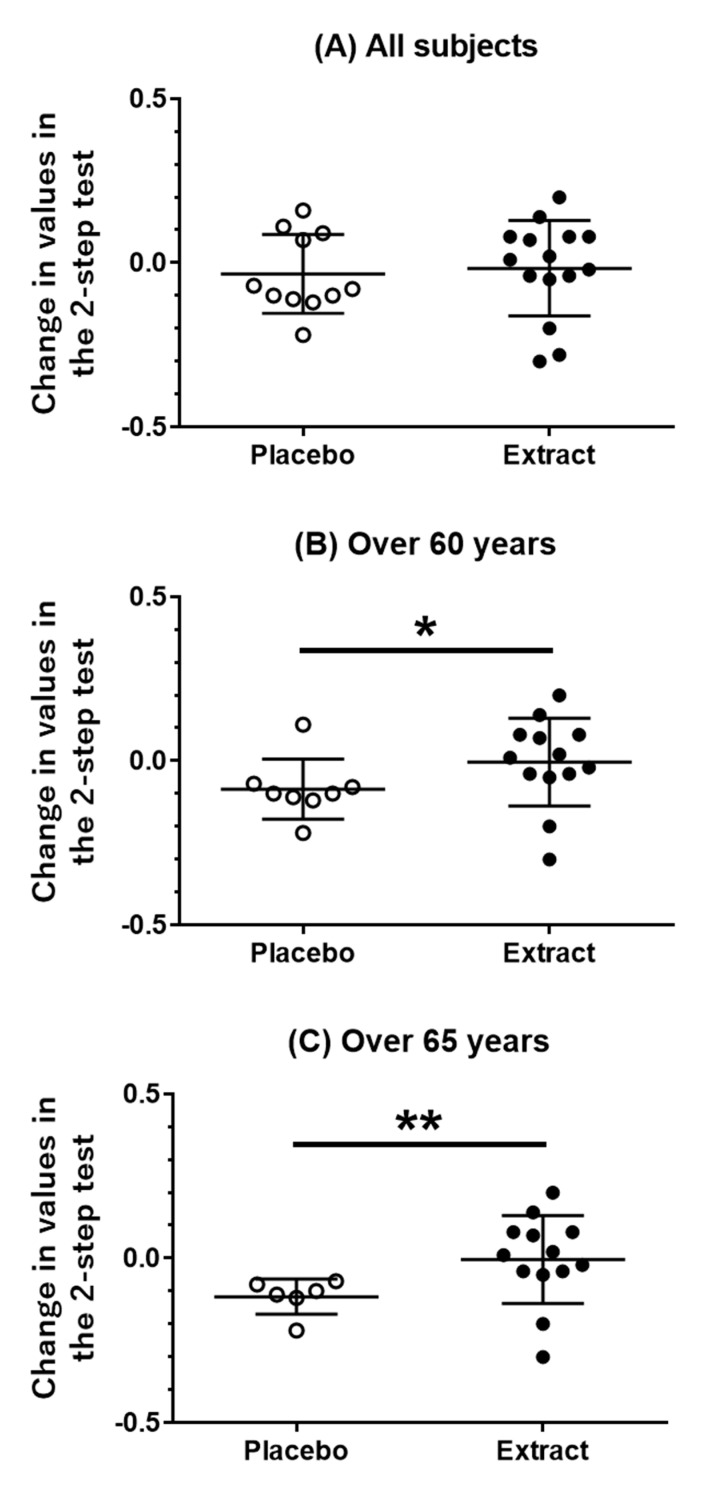
Change in values obtained in the two-step test in the placebo and CT extract groups. Change in values obtained in the two-step test pre-and post-treatment was compared. (**A**) All subjects (placebo; *n* = 11, CT extract; *n* = 15), (**B**) subjects aged > 60 years (placebo; *n* = 8, CT extract; *n* = 13), and (**C**) subjects aged > 65 years (placebo; *n* = 6, CT extract; *n* = 13) were analyzed. * *p* < 0.05, ** *p* < 0.01 (Mann Whitney test).

**Figure 3 nutrients-13-00264-f003:**
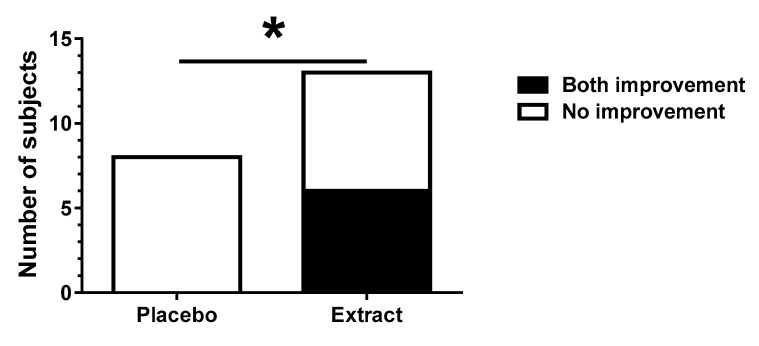
Number of subjects aged >60 years with improvements in the 5 m walking test and two-step test scores (placebo; *n* = 8, CT extract; *n* = 13). * *p* <0.05 (Fisher’s exact test).

**Table 1 nutrients-13-00264-t001:** Success score of the stand-up test.

	**Both Legs**			
	**40 cm**	**30 cm**	**20 cm**	**10 cm**
Success score	1	2	3	4
	**Right Legs**			
	**40 cm**	**30 cm**	**20 cm**	**10 cm**
Success score	5	10	15	20
	**Left Legs**			
	**40 cm**	**30 cm**	**20 cm**	**10 cm**
Success score	5	10	15	20

**Table 2 nutrients-13-00264-t002:** Sociodemographic and baseline characteristics of the study population.

Item	Placebo (*n* = 11)			Extract (*n* = 15)			*p* Value	
Male, *n*	2 (18.2%)			5 (33.3%)				
Age (years)	61.3	±	8.8	69.2	±	7.7	0.01	*
Height (cm)	156.5	±	8.3	155.6	±	9.1	0.83	
Body weight (kg)	61.3	±	13.4	56.9	±	8.1	0.44	
BMI (kg/m^2^)	24.9	±	4.1	23.5	±	2.8	0.39	
Locomo 7 score	2.1	±	1.2	2.0	±	1.8	0.50	

Values are given as male number and means ± SD. BMI, body mass index; * *p* < 0.05.

**Table 3 nutrients-13-00264-t003:** Comparison of muscle mass and body weight in all subjects.

All Subjects (*n* = 26)		Pre		Post		Intragroup Comparison (Pre vs. Post)				Changed Value (Post-Pre)		Comparison between Group (Extract vs. Placebo)			
Muscle Mass		Mean	SD	Mean	SD	95% CI		*p* Value	*r*	Mean	SD	95% CI		*p* Value	*r*
						Lower	Upper					Lower	Upper		
Whole body (kg)	Extract (*n* = 15)	38.63	6.73	38.58	6.65	−0.35	0.25	0.75	0.06	−0.05	0.51	−0.20	0.70	0.24	0.24
	Placebo (*n* = 11)	39.52	8.73	39.66	9.03	−0.35	0.60	0.40	0.17	0.15	0.67				
Body trunk(kg)	Extract	21.29	3.45	21.41	3.37	−0.25	0.45	0.83	0.04	0.13	0.72	−0.50	0.50	0.88	0.03
	Placebo	21.93	4.69	21.99	4.75	−0.30	0.40	0.84	0.04	0.06	0.57				
Arm right (kg)	Extract	2.00	0.45	2.00	0.42	−0.05	0.05	1.00	0.00	0.00	0.10	−0.10	0.10	0.80	0.05
	Placebo	1.97	0.50	1.98	0.52	−0.05	0.05	0.71	0.07	0.01	0.08				
Arm left (kg)	Extract	1.94	0.44	1.92	0.42	−0.10	0.05	0.59	0.11	−0.02	0.12	0.00	0.10	0.22	0.26
	Placebo	1.89	0.54	1.94	0.55	0.00	0.10	0.13	0.30	0.05	0.09				
Leg right (kg)	Extract	6.76	1.43	6.66	1.40	−0.30	0.10	0.43	0.16	−0.10	0.32	−0.10	0.20	0.81	0.08
	Placebo	6.86	1.60	6.85	1.67	−0.10	0.10	0.90	0.02	−0.01	0.15				
Leg left (kg)	Extract	6.65	1.46	6.59	1.45	−0.20	0.10	0.70	0.08	−0.06	0.31	−0.10	0.30	0.44	0.16
	Placebo	6.86	1.63	6.90	1.74	−0.10	0.20	0.57	0.11	0.04	0.22				
Body weight (kg)	Extract	56.91	8.09	57.06	8.39	−0.85	0.95	0.84	0.04	0.15	1.78	−1.20	0.60	0.33	0.19
	Placebo	61.35	13.40	61.00	13.61	−0.80	0.15	0.13	0.30	−0.35	0.69				

**Table 4 nutrients-13-00264-t004:** Comparison of physical activity scores in all subjects.

All Subjects (*n* = 26)		Pre		Post		Intragroup Comparison (Pre vs. Post)				Changed Value (Post-Pre)		Comparison between Group (Extract vs. Placebo)			
Physical Activity		Mean	SD	Mean	SD	95% CI		*p* Value	*r*	Mean	SD	95% CI		*p* Value	*r*
						Lower	Upper					Lower	Upper		
Hand grip right (kg)	Extract (*n* = 15)	26.70	9.63	27.47	9.28	−1.00	2.75	0.33	0.19	0.77	4.24	−1.50	3.00	0.47	0.14
	Placebo (*n* = 11)	29.95	7.29	31.68	7.19	0.50	3.00	0.04	0.40	1.73	3.45				
Hand grip left (kg)	Extract	25.62	8.71	26.67	8.89	−0.75	2.50	0.27	0.22	1.05	2.92	−2.50	2.00	0.88	0.03
	Placebo	28.09	8.63	29.45	7.72	−1.00	5.00	0.44	0.15	1.36	4.15				
5-m walking (s)	Extract	3.09	1.63	2.88	1.49	−0.34	0.03	0.13	0.30	−0.21	0.51	−0.19	0.35	0.47	0.14
	Placebo	2.33	0.33	2.26	0.36	−0.29	0.15	0.37	0.17	−0.07	0.30				
2-step test (point)	Extract	1.36	0.23	1.34	0.19	−0.11	0.07	1.00	0.00	−0.02	0.14	−0.15	0.10	0.54	0.13
	Placebo	1.43	0.19	1.40	0.17	−0.11	0.04	0.42	0.16	−0.03	0.12				
Stand-up test (point)	Extract	12.73	9.95	14.20	7.81	0.00	5.00	0.36	0.18	1.47	7.77	−5.00	4.00	0.76	0.06
	Placebo	14.45	11.86	15.91	12.23	−3.00	5.50	0.78	0.06	1.45	7.57				
GLFS-25 (point)	Extract	13.53	15.36	13.33	10.42	−4.00	0.00	0.68	0.08	−0.20	11.99	−9.00	2.00	0.22	0.25
	Placebo	11.73	13.54	9.91	13.04	−4.00	0.00	0.20	0.25	−1.82	3.82				

**Table 5 nutrients-13-00264-t005:** Comparison of physical activity scores in subjects aged >60 years.

Over 60 Years (*n* = 21)		Pre		Post		Intragroup Comparison (Pre vs. Post)				Changed Value (Post-Pre)		Comparison between Group (Extract vs. Placebo)				
Physical Activity		Mean	SD	Mean	SD	95% CI		*p* Value	*r*	Mean	SD	95% CI		*p* Value		*r*
						Lower	Upper					Lower	Upper			
Hand grip right (kg)	Extract (*n* = 13)	10.40	2.88	9.99	2.77	−1.75	2.75	0.61	0.10	0.50	4.47	−2.00	3.50	0.46		0.16
	Placebo (*n* = 8)	28.63	7.28	30.38	7.29	−1.50	5.75	0.12	0.30	1.75	4.06					
Hand grip left (kg)	Extract	9.34	2.59	9.48	2.63	−1.00	3.25	0.23	0.23	1.21	3.10	−3.50	2.50	0.92		0.02
	Placebo	26.50	8.93	27.88	7.20	−2.25	5.25	0.87	0.03	1.38	4.89					
5-m walking (s)	Extract	1.73	0.48	1.57	0.44	−0.36	0.06	0.24	0.23	−0.20	0.55	−0.29	0.39	0.50		0.14
	Placebo	2.24	0.19	2.18	0.24	−0.29	0.20	0.48	0.14	−0.07	0.30					
2-step test (point)	Extract	0.23	0.07	0.20	0.06	−0.09	0.08	0.86	0.03	−0.01	0.13	−0.18	−0.02	0.05	*	0.40
	Placebo	1.49	0.11	1.40	0.12	−0.16	0.00	0.09	0.33	−0.09	0.09					
Stand-up test (point)	Extract	12.38	10.50	13.31	6.93	−2.50	5.00	0.55	0.12	0.92	7.99	−5.00	7.00	0.86		0.04
	Placebo	14.00	11.83	17.00	12.10	−2.50	9.50	0.46	0.15	3.00	7.75					
GLFS-25 (point)	Extract	16.55	4.59	10.89	3.02	−9.50	7.50	0.64	0.09	−0.08	12.78	−9.00	3.00	0.34		0.20
	Placebo	8.00	3.85	7.00	5.95	−4.00	2.00	0.50	0.13	−1.00	3.78					

SD, standard deviation; * *p* < 0.05.

**Table 6 nutrients-13-00264-t006:** Comparison of physical activity scores in subjects aged >65 years.

Over 65 Years (*n* = 19)		Pre		Post		Intragroup Comparison (Pre vs. Post)					Changed Value (Post-Pre)		Comparison between Group (Extract vs. Placebo)				
Physical Activity		Mean	SD	Mean	SD	95% CI		*p* Value		*r*	Mean	SD	95% CI		*p* Value		*r*
						Lower	Upper						Lower	Upper			
Hand grip right (kg)	Extract (*n* = 13)	10.40	2.88	9.99	2.77	−1.75	2.75	0.61		0.10	0.50	4.47	−3.00	6.00	0.58		0.12
	Placebo (*n* = 6)	29.92	7.95	31.83	7.79	−1.75	6.75	0.25		0.23	1.92	4.79					
Hand grip left (kg)	Extract	9.34	2.59	9.48	2.63	−1.00	3.25	0.23		0.23	1.21	3.10	−4.00	7.00	1.00		0.01
	Placebo	27.08	10.34	28.83	8.03	−3.00	8.75	0.92		0.02	1.75	5.72					
5-m walking (s)	Extract	1.73	0.48	1.57	0.44	−0.36	0.06	0.24		0.23	−0.20	0.55	−0.33	0.35	0.52		0.13
	Placebo	2.28	0.20	2.20	0.27	−0.33	0.22	0.35		0.19	−0.09	0.32					
2-step test (point)	Extract	0.23	0.07	0.20	0.06	−0.09	0.08	0.86		0.03	−0.01	0.13	−0.21	−0.04	0.01	**	0.48
	Placebo	1.53	0.10	1.41	0.13	−0.17	−0.08	0.03	*	0.43	−0.12	0.05					
Stand-up test (point)	Extract	12.38	10.50	13.31	6.93	−2.50	5.00	0.55		0.12	0.92	7.99	−5.00	9.00	1.00		0.00
	Placebo	14.83	12.80	17.67	13.88	−5.00	11.50	0.89		0.03	2.83	8.89					
GLFS-25 (point)	Extract	16.55	4.59	10.89	3.02	−9.50	7.50	0.64		0.09	−0.08	12.78	−11.00	5.00	0.42		0.17
	Placebo	8.83	4.17	8.00	6.66	−4.50	3.50	0.72		0.07	−0.83	4.36					

SD, standard deviation; * *p* < 0.05, ** *p* < 0.01.

**Table 7 nutrients-13-00264-t007:** Changes in evaluated blood parameters following the intake of CT extract and placebo.

	Changed Values				Between Groups			
	Placebo		Extract		*p* Value	95% CI		
	Mean	SD	Mean	SD				
HDL-cholesterol	−3.27	5.76	−1.73	7.47	0.57	−4.04	to	7.12
Total protein	−0.01	0.24	−0.01	0.23	0.96	−0.20	to	0.19
Billirubin direct	0.01	0.03	0.01	0.03	0.83	−0.03	to	0.02
Billirubin indirect	0.01	0.07	0.05	0.12	0.38	−0.05	to	0.12
Glucose	−2.18	17.71	0.33	13.79	0.69	−10.22	to	15.25
Total cholesterol	−1.00	12.30	4.33	15.46	0.35	−6.33	to	16.99
Triglyceride	13.91	72.02	53.27	166.34	0.47	−71.48	to	150.20
Urea nitrogen	0.95	2.37	0.07	3.10	0.43	−3.19	to	1.42
Creatinine	0.02	0.03	0.01	0.11	0.72	−0.09	to	0.06
Uric acid	−0.03	0.40	0.01	0.63	0.88	−0.41	to	0.48
Na	1.18	1.25	0.00	1.00	0.01 *	−2.09	to	−0.27
K	0.03	0.36	−0.01	0.29	0.79	−0.30	to	0.23
Cl	0.36	2.25	−0.33	1.11	0.31	−2.08	to	0.68
Amylase	−11.64	37.07	−0.67	5.18	0.27	−8.90	to	30.84
Creatine kinase	−3.82	16.68	−0.13	24.15	0.67	−13.81	to	21.18
Leucine aminopeptidase	5.27	8.82	−0.47	4.03	0.04 *	−11.04	to	−0.44
γ-GTP	4.00	9.07	−0.53	5.48	0.13	−10.43	to	1.36
Cholinesterase	−1.00	26.80	2.47	28.54	0.76	−19.33	to	26.27
AST(GOT)	1.55	4.27	−1.27	4.03	0.10	−6.20	to	0.57
ALT(GPT)	1.91	5.05	−0.07	6.92	0.43	−7.06	to	3.11
Lactate dehydorogenase	−0.36	20.25	−6.33	34.14	0.61	−29.87	to	17.93
Alkaline phosphatase	−11.00	30.41	12.87	68.34	0.29	−21.82	to	69.55
Hemoglobin	−0.08	0.56	0.00	0.49	0.70	−0.34	to	0.51
Erythrocytes	−1.45	24.86	−0.07	18.48	0.87	−16.12	to	18.90
Leukocytes	−300.00	734.85	−280.00	838.54	0.95	−632.90	to	672.90
Hematocrit	−0.64	2.35	−0.47	1.59	0.83	−1.43	to	1.76
Erythrocyte mean corpuscular volume	−1.12	0.94	−1.17	1.55	0.92	−1.15	to	1.04
Erythrocyte mean corpuscular hemoglobin	−0.12	0.47	−0.03	0.64	0.71	−0.39	to	0.56
Erythrocyte mean corpuscular hemoglobin concentration	0.27	0.55	0.36	0.46	0.66	−0.32	to	0.50
Platelet	−0.82	2.70	−0.82	2.57	1.00	−2.15	to	2.15
LDL-cholesterol	−0.18	11.36	1.20	14.09	0.79	−9.29	to	12.05
Albumin	0.05	0.15	0.02	0.25	0.77	−0.20	to	0.15
Albumin	0.75	1.45	0.47	1.79	0.67	−1.64	to	1.07
A1-G	−0.04	0.21	−0.20	0.37	0.20	−0.42	to	0.09
A2-G	−0.19	0.43	−0.15	0.62	0.86	−0.41	to	0.49
B1-G	−0.18	0.31	−0.17	0.45	0.96	−0.32	to	0.33
B2-G	−0.07	0.22	−0.07	0.60	1.00	−0.39	to	0.39
G-G	−0.27	0.85	0.13	0.56	0.15	−0.16	to	0.97
A/G	0.05	0.09	0.05	0.09	0.97	−0.07	to	0.08

HDL-cholesterol, High-density lipoprotein cholesterol; Na, natrium; K, kalium; Cl, chlorine; γ-GTP, γ-glutamyl transpeptidase; AST(GOT), asparate aminotranferase glutamic oxaloacetic transaminase; ALT(GPT), alanine aminotransferase glutamic pyruvic transaminase; LDL-cholesterol, low density lipoprotein cholesterol; SD, standard deviation; * *p* < 0.05.

## Data Availability

The data presented in this study are available on request from the corresponding author.

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
