# Peer review of "Effects of Cistanche tubulosa Wight Extract on Locomotive Syndrome: A Placebo-Controlled, Randomized, Double-Blind Study"

_nutrients, 2021, doi:10.3390/nu13010264_

Round 1
Reviewer 1 Report
Congratulations.
This is very good paper.
Author Response
Thank you very much for reviewing our paper.
Reviewer 2 Report
It is an interesting study, corcerning a small number of cases, but a correct methodology and a randomised design.
The argument is important: in the discussion you might cite an article demonstrating that the effect of exercise and essential aminoacid can be additive, together with the citation n.22, that does not demonstrate this positive interaction.
- Cell Metabolism 12 362-372 October 6, 2010
Branched-ChainAminoAcidSupplementationPromotes
Survival and Supports Cardiac and Skeletal Muscle
Mitochondrial Biogenesis inMiddle-Aged Mice
Giuseppe D’Antona,1,2 Maurizio Ragni,3 Annalisa Cardile,3 Laura Tedesco,3,4 Marta Dossena,3,5 Flavia Bruttini,1,2
Francesca Caliaro,1,2 Giovanni Corsetti,6 Roberto Bottinelli,1,2 Michele O. Carruba,3,4 Alessandra Valerio,3,5
and Enzo Nisoli3,4,*
Author Response
We thank the editors and reviewers for their positive comments, thorough evaluation of our manuscript, and for the opportunity to improve our paper. We provide here a point-by-point response to the reviewers' comments that we believe thoroughly address their concerns. The revised parts are shown in yellow in the main text.
Point : The argument is important: in the discussion you might cite an article demonstrating that the effect of exercise and essential aminoacid can be additive, together with the citation n.22, that does not demonstrate this positive interaction.
Cell Metabolism 12 362-372 October 6, 2010
Branched-ChainAmino AcidSupplementationPromotes Survival and Supports Cardiac and Skeletal Muscle Mitochondrial Biogenesis in Middle-Aged Mice
Giuseppe D'Antona, 1,2 Maurizio Ragni,3 Annalisa Cardile,3 Laura Tedesco,3,4 Marta Dossena,3,5 Flavia Bruttini,1,2 Francesca Caliaro, 1,2 Giovanni Corsetti,6 Roberto Bottinelli, 1,2 Michele O. Carruba,3,4 Alessandra Valerio,3,5 and Enzo Nisoli3,4,*
Response : As you pointed out, I cited the article in the discussion.(l.287-288)

Reviewer 3 Report
The objective of this study was to examine the effects of Cistanche Tubulosa on locomotive syndrome
Introduction
Overall you did a good job with the introduction, here are some specific comments
- You state that studies that examine ways to improve locomotive syndrome tend to focus on pre-post exercise in the same groups and do not include a control group. You need to cite this statement.
- You state that there are a few interventional studies that use a placebo-controlled, randomized, double-blinded trial however you don't cite any. Please cite the studies that do utilize this methodology.
- In your introduction you should state that you conducted this trial in humans. In the third paragraph you talk about using it in mice however, your actual study in conducted in human adults.
Methodology
Overall this was a well written methodology. Below are my specific comments
- You should not talk about the number of subjects and such in your study design. Instead you should just focus on what your study design was, and talk about the intervention (e.g. 1800m.g. per day) and placebo. Your description of the products in the production section is fine however, there should be some mention of it in your trial design. Additionally, in this section you should mention the length of time of your intervention.
- How were participants recruited? How was screening performed? Who performed the screening?
- Handgrip strength does not fall under muscle mass. You need to have a separate measurement for handgrip strength.
- You should mention in your muscle mass section that you're using a segmental BIA and you should report the validity and reliability of this piece of equipment as well.
- Where were the 5m walking tests performed? How were they performed? How was time recorded? Did they walk from cone to cone? Was it a human collecting the speed? Please provide more detail on the 5m test
- How was the 2-step test performed? What were the instructions given to the participants? What comprises of the measurement? (e.g. is it from the toes when the participant is standing to the heel of the final step? Or is it to the toes of the final step?)
- Please provide the Cronbach's alphas for the GLFS-25 from previous literature and your present study.
- Did you conduct an a prior power analysis?
- In your statistical analysis section you mention using non-parametric tests only. I'm assuming that's because the data was not normally distributed. Please provide the reader with the techniques you used to test for normality of distribution of your outcome variables. Additionally, provide the reader with any information of transformation techniques you might have used with the data to try to normalize it. Also please indicate if data has similar skewness or kurtosis (e.g. GLFS-25 scores were positively skewed for both groups).
- Please provide a detailed procedure section in which you describe how the participants were provided the interventions (whether it was placebo or CT), how and where were the tests performed? What was the order in which the tests were performed on testing day? What was the rest time between each test?
- In your limitations section you state that you did not control for daily diet or physical activity. Did you measure it at all? At baseline or when participants were tested?
Results
Overall you did a good job of presenting the results. Below are my comments
- Table 2 is not very intuitive. I am assuming that the second row is sex. Which ones are male/female? Also is there a significant difference in gender composition between groups. Also I am assuming cm stands for height (the numbers are way too big to suggest otherwise). It could just be that copying the table cut certain parts.
- You repeat the fact that participants in the CT group were older in the initial part of your results section.
- You compared the number of subjects who's scores improved for 5m walk and 2-step test. Did you do the same for the other parameters? If not, then explain why
Discussion
Overall, I think the discussion was good. I do believe after you have addressed some of the methodological questions you may want to add to your limitations section.
Author Response
We thank the editors and reviewers for their positive comments, thorough evaluation of our manuscript, and for the opportunity to improve our paper. We provide here a point-by-point response to the reviewers' comments that we believe thoroughly address their concerns. The revised parts are shown in yellow in the main text.
Introduction
Point 1 : You state that studies that examine ways to improve locomotive syndrome tend to focus on pre-post exercise in the same groups and do not include a control group. You need to cite this statement.
Response 1 : We added the cited references that you pointed out. (l.48)
Point 2 : You state that there are a few interventional studies that use a placebo-controlled, randomized, double-blinded trial however you don't cite any. Please cite the studies that do utilize this methodology.
Response 2 : We added new references 5.
[5] Suzukamo, C.; Ishimaru, K.; Ochiai, R.; Osaki, N.; Kato, T. Milk-fat globule membrane plus glucosamine improves joint function and physical performance: a randomized, double-blind, placebo-controlled, parallel-group study. J. Nutr. Sci. Vitaminol (Tokyo). 2019, 65(3),242-250. doi: 10.3177/jnsv.65.242.
Point 3 : In your introduction you should state that you conducted this trial in humans. In the third paragraph you talk about using it in mice however, your actual study in conducted in human adults.
Response 3 : We added a note at the end of the introduction that this is a human study. (l.64-65)
Materials and Methods
Point 1 : You should not talk about the number of subjects and such in your study design. Instead you should just focus on what your study design was, and talk about the intervention (e.g. 1800m.g. per day) and placebo. Your description of the products in the production section is fine however, there should be some mention of it in your trial design. Additionally, in this section you should mention the length of time of your intervention.
Response 1 : The study design, subject information, and interventions were revised. (l.68-109)
Point 2 : How were participants recruited? How was screening performed? Who performed the screening?
Response 2 : We added a note on 2.1. Trial Design. The screening was conducted by the participants themselves, and they were asked to confirm the contents of the locomotive check (Locomo 7) when they apply. (L.69-70)
Point 3 : Handgrip strength does not fall under muscle mass. You need to have a separate measurement for handgrip strength.
Response 3 : As you pointed out, grip strength is not muscle mass, so the item has been listed separately. (l.133-135)
Point 4 : You should mention in your muscle mass section that you're using a segmental BIA and you should report the validity and reliability of this piece of equipment as well.
Response 4 : The MC-780A body composition analyzer is a device used in Japanese government statistics and has been used in several academic studies, so we believe that its reliability and validity are maintained.
Point 5 : Where were the 5m walking tests performed? How were they performed? How was time recorded? Did they walk from cone to cone? Was it a human collecting the speed? Please provide more detail on the 5m test
Response 5 : We revised 2.4.3. Five-meter walking test as follows; “Five-meter walking speed” measures the time that passes during a 5 m walking section by setting acceleration sections of 1.0 m at the start of a total 6 m of walking length. A start point, 1.0 m and finish points were marked by lines. A date collector judged and measured time using a stop watch. Measurements were performed twice, and the fastest walking speed, without running, was recorded.
Point 6 : How was the 2-step test performed? What were the instructions given to the participants? What comprises of the measurement? (e.g. is it from the toes when the participant is standing to the heel of the final step? Or is it to the toes of the final step?)
Response 6 : We revised 2.4.3. Five-meter walking test as follows; “The two-step test assesses walking ability [12]. The subject starts from a standing posture and is asked to take steps forward with maximum stride without losing balance. A specific mat with scales for the two-step test (JOA) was used. The length of two steps was measured from toe to toe. Before performing the test, an instructor demonstrated stepping. The score is calculated using the total length of two steps (cm) divided by the subject's height (cm).”
Point 7 : Please provide the Cronbach's alphas for the GLFS-25 from previous literature and your present study.
Response 7 : The Cronbach’s alpha of GLFS-25 in the previous study was 0.959(Chaochen et al., PeerJ, 8: e9026, 2020), and that of in this study was 0.962.
Point 8 : Did you conduct an a prior power analysis?
Response 8 : The prior power analysis was conducted. Calculated using the Wilcoxon test with an effect size of 0.5, significance level of 0.05, and power of 0.8, and then the required sample size was 28, which was considered to be sufficient because 32 participants had been determined at the recruit phase. We added the description about that in Materials and Methods. (l.76-79)
Point 9 : In your statistical analysis section you mention using non-parametric tests only. I'm assuming that's because the data was not normally distributed. Please provide the reader with the techniques you used to test for normality of distribution of your outcome variables. Additionally, provide the reader with any information of transformation techniques you might have used with the data to try to normalize it. Also please indicate if data has similar skewness or kurtosis (e.g. GLFS-25 scores were positively skewed for both groups).
Response 9 : Two tests of normality (Kolmogorov-Smirnov test and Shapiro-Wilk test) were conducted in SPSS; since two tests showed no normality, a nonparametric test was employed for all score analyzed.
Point 10 : Please provide a detailed procedure section in which you describe how the participants were provided the interventions (whether it was placebo or CT), how and where were the tests performed? What was the order in which the tests were performed on testing day? What was the rest time between each test?
Response 10 : The description about intervention was added to the assessment section.(l.123-127)
Point 11 : In your limitations section you state that you did not control for daily diet or physical activity. Did you measure it at all? At baseline or when participants were tested?
Response 11 : Subjects were told not to make any extreme changes to their diet or exercise habits while participating in the study, but they did not confirm how this actually worked out. Since we did not control for diet and exercise in this study, we would like to confirm this in future studies.
Result
Point 1 : Table 2 is not very intuitive. I am assuming that the second row is sex. Which ones are male/female? Also is there a significant difference in gender composition between groups. Also I am assuming cm stands for height (the numbers are way too big to suggest otherwise). It could just be that copying the table cut certain parts.
Response 1 : Units and other information were added, and Table 2 was revised. (l.201)
Point 2 : You repeat the fact that participants in the CT group were older in the initial part of your results section.
Response 2 : One of duplicate descriptions were deleted.
Point 3 : You compared the number of subjects who's scores improved for 5m walk and 2-step test. Did you do the same for the other parameters? If not, then explain why
Response 3 : Similar comparisons were analyzed all other items, but no significant differences were found in other combination, so they were not included in the results.

Reviewer 4 Report
The aim of the present study was to evaluate the effects of C. tubulosa (CT) extract on the locomotive syndrome.
The authors concluded that this placebo-controlled, randomized, double-blind study demonstrated that treatment with CT extract significantly prevented a decline in walking ability without any notable adverse effects in patients with locomotive syndrome.
The manuscript is well written and present interesting findings. However, some modifications are required:
I suggest adding some statistical value in the results of the abstract.
Add some hypothesis at the end of the Introduction.
Did the authors calculate the sample size.
Add the confidence interval and the effect size in the statistical analysis and the results.
Add the unit of measurement for all parameters in the tables.
Add some practical recommendations.
Author Response
We thank the editors and reviewers for their positive comments, thorough evaluation of our manuscript, and for the opportunity to improve our paper. We provide here a point-by-point response to the reviewers' comments that we believe thoroughly address their concerns. The revised parts are shown in yellow in the main text.
Point 1 : I suggest adding some statistical value in the results of the abstract.
Response 1 : Statistics were included in the abstract.
Point 2 : Add some hypothesis at the end of the Introduction.
Response 2 : Hypothesis was added to the last sentence of the introduction.(l.65)
Point 3 : Did the authors calculate the sample size.
Response 3 : The power of the test was calculated in advance. For the Wilcoxon test, calculated with an effect size of 0.5, significance level of 0.05, and power of 0.8, the required sample size was 28, which was considered to be sufficient since 32 participants had been determined at the time of participant determination. We added the description about that in Materials and Methods. (l.76-79)
Point 4 : Add the confidence interval and the effect size in the statistical analysis and the results.
Response 4 : The effect size is shown in the Table 3-6.
Point 5 : Add the unit of measurement for all parameters in the tables.
Response 5 : Units were added to Table 3-6.
Point 6 : Add some practical recommendations
Response 6 : We added future application in conclusions as follows; “Based on the evidence in this study, we will promote the development of CT extract as Foods with Functional Claims and Over-The-Counter medicine in Japan.” (l.309-311)

Round 2
Reviewer 3 Report
Thank you very much for addressing most of my concerns.
One issue I did have with what you addressed is that you did not describe your distribution of data to justify the non-parametric tests in your statistical analysis section.
Now that the tables and results are more intuitive I have another couple of suggestions.
In the discussion section you need to address the fact that the participants in your experimental condition were significantly older which may have contributed to some of the non-significant findings. It is plausible that they may have had greater age associated declines which could have led to your results being non-significant.
I think you should also mention the fact that it is significant that despite the large age difference the experimental group reported significant improvements in gait speed and step width.
Other than that you did a good job of addressing my concerns.
Author Response
We thank the editors and reviewers for their positive comments, thorough evaluation of our manuscript, and for the opportunity to improve our paper. We provide here a point-by-point response to the reviewers' comments that we believe thoroughly address their concerns. The revised parts are shown in yellow in the main text.
Point 1:One issue I did have with what you addressed is that you did not describe your distribution of data to justify the non-parametric tests in your statistical analysis section.
Response1:We added in Statistics as follows; “Based on the results of the two tests of normality (Kolmogorov-Smirnov test and Shapiro-Wilk test) and the values of skewness and kurtosis, it was decided to use nonparametric tests for all statistical analyses.”
Point 2 : In the discussion section you need to address the fact that the participants in your experimental condition were significantly older which may have contributed to some of the non-significant findings. It is plausible that they may have had greater age associated declines which could have led to your results being non-significant.
I think you should also mention the fact that it is significant that despite the large age difference the experimental group reported significant improvements in gait speed and step width.
Response 2 : We added in Discussion as follows; “Despite the age of CT group was older than that of placebo group (Table2), CT intake significantly improved those physical activities. This result indicates the effect of CT extract is probably reliable.”
